# Parkinson’s Disease Patient Monitoring: A Real-Time Tracking and Tremor Detection System Based on Magnetic Measurements ^†^

**DOI:** 10.3390/s21124196

**Published:** 2021-06-18

**Authors:** Filippo Milano, Gianni Cerro, Francesco Santoni, Alessio De Angelis, Gianfranco Miele, Angelo Rodio, Antonio Moschitta, Luigi Ferrigno, Paolo Carbone

**Affiliations:** 1Department of Electrical and Information Engineering, University of Cassino and Southern Lazio, 03043 Cassino, Italy; filippo.milano@unicas.it (F.M.); g.miele@unicas.it (G.M.); ferrigno@unicas.it (L.F.); 2Department of Medicine and Health Sciences “Vincenzo Tiberio”, University of Molise, 86100 Campobasso, Italy; gianni.cerro@unimol.it; 3Department of Engineering, University of Perugia, 06125 Perugia, Italy; francesco.santoni@unipg.it (F.S.); alessio.deangelis@unipg.it (A.D.A.); paolo.carbone@unipg.it (P.C.); 4Department of Human, Social and Health Sciences, University of Cassino and Southern Lazio, 03043 Cassino, Italy; a.rodio@unicas.it

**Keywords:** Parkinson’s disease, magnetic positioning, measurement system, tremor classification

## Abstract

Reliable diagnosis of early-stage Parkinson’s disease is an important task, since it permits the administration of a timely treatment, slowing the progression of the disease. Together with non-motor symptoms, other important signs of disease can be retrieved from the measurement of the movement trajectory and from tremor appearances. To measure these signs, the paper proposes a magnetic tracking system able to collect information about translational and vibrational movements in a spatial cubic domain, using a low-cost, low-power and highly accurate solution. These features allow the usage of the proposed technology to realize a portable monitoring system, that may be operated at home or in general practices, enabling telemedicine and preventing saturation of large neurological centers. Validation is based on three tests: movement trajectory tracking, a rest tremor test and a finger tapping test. These tests are considered in the Unified Parkinson’s Disease Rating Scale and are provided as case studies to prove the system’s capabilities to track and detect tremor frequencies. In the case of the tapping test, a preliminary classification scheme is also proposed to discriminate between healthy and ill patients. No human patients are involved in the tests, and most cases are emulated by means of a robotic arm, suitably driven to perform required tasks. Tapping test results show a classification accuracy of about 93% using a k-NN classification algorithm, while imposed tremor frequencies have been correctly detected by the system in the other two tests.

## 1. Introduction

In 2019, the United Nations released a world population prospect [1] that certifies how the population is increasing in terms of both size and age. The year 2018 was the first year when the number of people over 65 exceeded the number of children under 5. Furthermore, the life expectancy of male Europeans increased from 69 years in 1990 to 81 years in 2018. This age increase is associated with a higher probability of developing neuropsychiatric neurodegenerative disorders [2]. Among them, Parkinson’s disease (PD) is the second most prevalent disorder [3], and its incidence exponentially grows with age [4], and several risk factors are reported in [5]. Affected patients show motor and non-motor symptoms, and quality of life is also progressively worsened by the disease. Early diagnosis of PD is crucial for prompt intervention with suitable pharmacological therapy, thus slowing the progression of the disease.

The diagnosis of PD can be made through the evaluation of motor symptoms after a specialist visit by a neurologist [6]. The neurologist observes the patient performing specific tasks, and then assigns a score to each test, based on the unified classification scale of the PD, or Unified Parkinson’s Disease Rating Scale (UPDRS). The Hoehn and Yahr (HY) scale is used to evaluate the pathological progression of the disease. The main problem with these methods is that the classification depends on strongly subjective factors. A study reveals that around 25% of diagnoses are incorrect, particularly when essential tremor and atypical parkinsonian symptoms occur [7]. To perform a complete analysis, measurements performed over several days must be executed and, in addition, clinical tests should often be accompanied by instrumental tests such as nuclear magnetic resonance, positron emission tomography (PET) and single photon emission tomography (SPECT) which allow for studying the functioning of the cells directly and identifying pathological processes. In this case, patients are forced to stay for several days in health facilities, increasing the costs incurred and the crowding of the facilities. Moreover, expensive instruments are required that involve advanced technologies available only in specialized centers.

For these reasons, the use of low-cost, low-power, unobtrusive and accurate sensor devices has been analyzed, aiming at diagnosing PD in its early stage. A typical task is the analysis of tremor, which is one of the preeminent symptoms and permits the evaluation of the progression of PD. Most of the proposed systems are based on wearable inertial measurement unit (IMU) devices [8,9,10,11]. These devices can monitor in real time physical features such as vibration and orientation and transmit acquired data directly to medical centers, where diagnoses can be performed. Another important diagnostic tool is represented by speech analysis [12,13] adopting machine learning techniques. Reviews such as [13,14,15] summarize the current state of the art in Parkinson’s diagnosis-assisting technologies.

Published technologies are generally focused on wearable devices, in order to give patients the possibility to perform activities of daily living, and therefore to analyze their degree of disease without being supervised by a doctor in a laboratory. They mainly use accelerometers, gyroscopes and surface electromyography, individually or jointly in some cases. From sensor data, it is possible to extract information on movement disorders such as bradykinesia, hypokinesia, dyskinesia and freezing of gait. Positions of wearable devices on the body must be optimized to find the most sensitive spots to capture meaningful data.

The detection is generally delegated to machine learning techniques which use captured data to train, validate and finally test for a specific disorder. Even if these solutions are very cheap, they can reliably estimate vibrations and orientation, but they often fail to correctly estimate the trajectories starting from acquired acceleration and orientation data, mainly because trajectories are determined by integration of inertial measurements, which is affected by accumulating errors and drift.

To overcome these limits, the authors of this paper propose the adoption of a scalable and transportable tracking and monitoring system. The solution presented in this work can be used to detect vibration patterns and trajectories compatible with those associated with PD symptoms and diagnostic tests, as well as for other localization and rehabilitation purposes, such as in [16,17,18]. The system measures vibrations from direct position measurements instead of integrating inertial data, avoiding error accumulation. Unlike state-of-the-art solutions, the primary aim of our system is to determine the 3D position and trajectory with a fast sampling rate, and to extract, from trajectory estimation, secondary data to detect possible PD symptoms. Such trajectory estimation is particularly important beyond simple tremor identification, as stated by UPDRS, in which several tests require trajectories, such as test 3.5 “Hand movement”, 3.6 “Pronation-supination movements of hands”, 3.8 “Leg agility”, etc. In all these tests, the speed, amplitude, hesitations, halts and decrementing amplitude must be evaluated with good uncertainties in repetitive movements. Artificial intelligence can be applied as well, but the system also allows analytical observation of PD tremor symptoms by a frequency-domain analysis. It is composed of a “sensorized box” which allows the patient to monitor his/her status by means of simple predetermined movements inside it, where the trajectory of his/her limb can be tracked. It is essentially a localization system but, thanks to the high accuracy and a fast orientation/position estimation algorithm, it is also capable of tracking tremor movements up to a frequency well beyond 10 Hz. In its first realization, it can estimate the position, the spatial orientation, and the trajectory of the hand in a 30 × 30 × 30 cm^3^ cube volume. It is based on low-power and high-frequency AC magnetic fields.

This study extends the results in [19], where the feasibility of adopting the magnetic tracking system for tremor monitoring was investigated by preliminary experiments, and it adopts a hardware setup which was entirely developed by our research group and presented in [16]. Extensive measurement results are provided on trajectories that emulate tests for movement tremor, rest tremor and tapping. Machine learning techniques are also applied to investigate the possibility of automatic classification. Therefore, a complete characterization of the proposed system for its potential application to Parkinson’s disease patient monitoring is provided. The paper is structured as follows: Section 2 describes the main tremor typologies associated with PD and available specific scientific solutions for their monitoring; Section 3 reports the developed setup; Section 4 provides experimental results with the help of an accurate robotic arm used to provide reference positions. Conclusions are then drawn in Section 5.

## 2. Description of the Analyzed Parkinson’s Symptoms

In 2018, the consensus statement (CS) on the classification of tremor [20] classified parkinsonian tremor among the forms of tremor associated with other pre-eminent neurological signs. The CS defined it as a rest tremor (RT), considered as the classic PD tremor, with a frequency of 4–7 Hz. According to several studies, RT is present in 70% of patients with PD at onset and during the disease it affects 75–100% of patients.

However, it has recently been reported that isolated RT (i.e., in the absence of other types of tremors) is present only in a small percentage (12–20%) of PD patients [20]. RT disappears with voluntary movement. Generally, it is asymmetrical and most commonly begins with the upper limb with movements that reproduce the act of “counting money”, and then spread to the ipsilateral lower limb. It can potentially also affect other sites in addition to the limbs, including the chin and jaw. Often, in the advanced stages of the disease, RT is present bilaterally but tends to maintain a certain asymmetry between the two sides. A recent study has shown that the only predictive factor for the spread of RT to initially unaffected body segments is the patient’s age [21]. According to this study, in patients with an onset age greater than 63 years, the spreading of tremor is more likely.

In PD, it is also possible to observe an action tremor (AT), which includes kinetic tremor (KT), postural tremor (PT) and intentional tremor (IT), with a frequency equal to or different from that of RT. KT occurs when a voluntary movement is performed, PT is a tremor that arises when the patient tries to maintain a position against gravity and IT occurs with goal-directed movement and worsens when approaching the target. KT has a characteristic frequency of about 5–8 Hz. PT results in a frequency of 5–8 Hz, while IT occurs at a frequency of 3–10 Hz. Although AT may affect a significant percentage of parkinsonian patients (46–93%), in 2015 the MDS Clinical Diagnostic Criteria for Parkinson’s Disease [22] established that the presence of isolated tremor (KT, PT or IT) is not a diagnostic criterion for PD. AT has a frequency of about 9 Hz, which is therefore significantly higher than that of RT. Many clinical studies have suggested that AT is a cause of major disability for parkinsonian patients because it interferes with voluntary movements.

The re-emerging tremor (ReT) occurs in patients who have RT, which disappears with the achievement of a posture and then appears a few seconds after the posture is maintained. The prevalence of ReT is around 25% according to clinical studies [23]. However, a recent neurophysiological study performed with electromyographic recordings of the flexor and extensor muscles of the upper limbs has suggested that 80% of patients with PT actually have ReT [24]. Although ReT occurs while taking a posture, it has been suggested that it represents a clinical variant of RT. Figure 1 shows a summary of the characteristic frequencies of the various tremors.

To evaluate the patient’s status or to preliminarily scout possible Parkinson’s precursors, two further tests are generally proposed by doctors: the stop movement (SM) tests and finger tapping (FT) tests. In the first case, the patient is required to perform a predefined trajectory, during which he/she must stop for some seconds and his/her capability to retain a fixed position is evaluated, along with the tremor associated with the rest and movement states. Concerning the tremor classification, we may classify tremors experienced during the SM test into two categories: intentional tremor during moving trajectory and postural tremor during stops (static position achieved).

The FT test is commonly used to assess bradykinesia, which is one of the principal symptoms associated with Parkinson’s disease [6]. The test is performed by asking a patient to open and close the thumb and index fingers as wide and as fast as possible. The FT test is included in the Unified Parkinson’s Disease Rating Scale defined by the Movement Disorder Society (MDS-UPDRS) [25]. Capturing finger movements, extracting features and comparing them between ill and healthy patients by using machine learning techniques is an important step towards the definition of criteria for the early diagnosis of Parkinson’s disease [26,27,28,29,30,31].

## 3. Description of the Magnetic Positioning System

In this section, we illustrate the application of a magnetic positioning system (MPS) with a high measurement rate (number of position and attitude measurements per second) to characterize frequencies and amplitudes of limb tremors. The high measurement rate allowed a real-time characterization of the tremor while a limb moved within the active volume of the MPS.

The MPS prototype used [16] was an enhanced version of that presented in [32], which was also previously adopted for industrial purposes in [33], and equipped with novel capabilities for reducing EMC disturbances in [17]. The system tracked the position and attitude of an active coil (TX) moving within a matrix of passive coils acting as receivers (RXs). According to Ampere’s equivalence principle, a coil traversed by an electric current is equivalent to a magnetic dipole. Hence, the application of an alternating current to TX produces an oscillating magnetic field. This field induces on each passive RX an electromotive force which is proportional to the time derivative of the flux of the magnetic field crossing the RX coil’s surface in accordance with Faraday’s principle. The induced voltages can be directly measured. The position and attitude of the TX can be estimated by inverting a suitable mathematical model based on the measured voltages.

The MPS included a matrix of 24 passive receivers, mounted on a carved wooden structure. Sixteen RXs were on the X–Y plane. The X–Z and Y–Z planes contained 4 RXs each. On the horizontal plane, 16 sensors were mounted on a regular 4 × 4 square grid of 24 cm on each side. At the center of both the vertical planes, 4 sensors were mounted on the vertices of an 8 cm side square. Even though a single plane configuration (X–Y only) was implemented in the literature, the tri-planar configuration was chosen for better positioning accuracy along all directions [34]. RX coils were in parallel with 680 nF capacitors to realize resonant LC circuits. Each RX coil had 252 windings with a radius of 9.5 mm. The capacitors were chosen in order to approximately match the resonance band of TX with that of each RX. The TX coil had 36 windings, a radius of 5 mm and it was connected in parallel with a 50 nF capacitor, thus forming a resonant LC circuit. Both TX and RX circuits were tuned at 182 kHz resonant frequency.

The magnetic field **B** generated by a single coil was modeled as an elementary magnetic dipole [34,35]. The root mean square (RMS) magnetic field at the center of the *i*_th_ receiving coil is:(1)$$B_{rms,i}=\frac{\mu_{0}}{4\pi }\frac{m_{tx}}{d_{i}^{3}}\left[3\left({\hat{n}}_{tx}\cdot {\hat{n}}_{d,i}\right){\hat{n}}_{d,i}-{\hat{n}}_{tx}\right]\;,\;i=1\ldots N$$
where $d_{i}$ is the distance between the transmitter and the *i*_th_ receiver, ${\hat{n}}_{d,i}$ is the unit vector associated with vector $d_{i}=r_{rx,i}-r_{tx}$ pointing from the transmitting to the receiving coil,$\;\mu_{0}$ is the vacuum magnetic permeability, $m_{tx}$ is the magnetic dipole moment of the TX coil ($m_{tx}=N_{tx}S_{tx}I$, where $N_{tx}$ is the number of windings, $S_{tx}$ is the surface of the coil and I is the current traversing it). By approximating B as homogeneous over the whole receiving coil surface, the induced RMS voltage on the *i*_th_ receiver is given by:(2)$$V_{rms,i}=2\pi f_{0}N_{rx}S_{rx}B_{rms,i}\cdot {\hat{n}}_{rx}\;,\;i=1\ldots N$$
where $N_{rx}$ is the number of windings of each receiving coil, $S_{rx}$ its surface and ${\hat{n}}_{rx}$ is the unit vector orthogonal to the coil, assuming a left-handed orientation of the coil. Writing position and attitude as a single vector $\theta ={\left[r_{tx},{\hat{n}}_{tx}\right]}^{T}$, the positioning problem is equivalent to minimizing the following cost function [36], considering N receivers:(3)$$F\left(\theta \right)=\overset{N}{\underset{i=1}{\sum }}{\left[{\tilde{V}}_{rms,i}-V_{rms,i}\left(\theta \right)\right]}^{2}$$
where ${\tilde{V}}_{rms,i}$ are the measured voltage values on each RX. The best estimate of θ is the value minimizing the cost function $\hat{\theta }=argmin\;F\left(\theta \right)$. As explained in [32], the optimization problem can be split into a linear and a non-linear part. This approach results in a very fast and stable computation, suitable for a real-time application. The optimization problem was solved using the Nelder–Mead algorithm.

The main features of the MPS—mean Euclidean error ${\bar{\sigma }}_{r}$, mean angular error ${\bar{\sigma }}_{\theta }$, measurement rate, mean signal-to-noise ratio $\bar{SNR}$ and TX operating frequencies—as reported in [16], are summarized in Table 1. We performed experiments using a robotic arm as a reference. A set of positions spaced 1 mm was realized using the robot, which featured a resolution of 0.1 mm. Our system was able to correctly detect 1 mm displacements. Therefore, the system resolution was 1 mm or better, with reference to the standard definition of the resolution as “the smallest change in a measurand that causes a perceptible change in the corresponding indication” [37]. Figure 2 shows the resulting distributions of measurement results for each position along the x coordinate. Each distribution was well separated from its neighbors. Even considering the worst case (positions 2 and 3), by performing a two-sample *t*-test, the hypothesis that the sample sets for both positions could possibly come from the same distribution could be rejected even when a very low significance level, such as 0.001, was considered.

## 4. Experimental Results

### 4.1. Experimental Setup

The basic idea was to steadily mount one or more TXs on the limb under scrutiny, so that the movements of a TX were rigidly linked with the limb tremor, thus using the MPS to capture limb movements plus tremor. The aim of this work was to validate this approach by tracking a TX which was moved along known and repeatable trajectories by a robotic arm, with a superimposed small oscillation, identified as tremor. The high MPS measurement rate allowed for a high-resolution trajectory tracking. Accordingly, the tremor components could be discriminated on the basis of their estimated frequencies and amplitudes.

The experimental setup is shown in Figure 3. The positioning system was calibrated through the procedure described in [32]. The calibration procedure was performed by moving the TX along known trajectories by means of the robotic arms. These reference trajectories provided the ground truth with respect to which all the parameters of the system were fixed. All parameters were adjusted in order to minimize the difference between ground truth and reconstructed trajectories.

The cut-out cardboard hand, reported in Figure 3a, is used for illustration purposes only. A schematic representation of the node positions is depicted in Figure 3b.

A servo motor was fixed at the end of the robotic arm. The hand was attached to the shaft of the servo motor. The tremor was obtained by alternately rotating the servo motor within a small angle (10°) so as to simulate limb tremor, at a nominal frequency that could be set up to 10 Hz. The servo motor was controlled by an Arduino UNO R3 microcontroller unit (MCU). The TX coil was attached on the middle finger. In previous work [19], an accelerometer was also fixed at the same point, in order to measure the same oscillation amplitude and frequency, so as to validate the MPS measure.

### 4.2. Tremor Test

The first test emulated an action tremor, specifically an intentional tremor, oscillating at 10 Hz. As described in Section 2, AT is defined as a tremor produced by voluntary muscle contraction, which occurs during any type of movement of an affected body part. The chosen trajectory is arbitrary, with the path configuration being irrelevant in PD tests. We used a square and a circle path as to test the system on both a smooth and a polygonal path. The TX was mounted on the index finger and the robotic arm moved the hand along a square and circular trajectory superimposing the 10 Hz tremor component through the use of a servo motor. AT was effective only along the x and y coordinates and no tremor along the z direction was applied, therefore, only the results for the x and y coordinates are shown and analyzed here. Nevertheless, the same theoretical approach would also be valid for the z coordinate. The duration of a complete trajectory took about 15 s and the test was repeated 10 times. Figure 4 shows the acquired trajectories for all of the 10 repetitive tests.

In the first step, in order to measure the tremor component, the original signal was filtered with a high-pass IIR eighth order filter with a cut-off frequency of 2.5 Hz. Then a discrete Fourier transform (DFT) with a rectangular window was applied to the filtered signal. Figure 5 shows a comparison between the spectrum of the original signal (not filtered) and the spectrum of the filtered signal.

The tremor component (10 Hz) introduced through the servomotor is clearly identifiable on both axes. The low-frequency components of the original signal (less than the cut-off frequency) were due to the imposed trajectory. In particular, the signal shown in Figure 5, representing the x and y trajectory coordinates, was normalized to its maximum value. In detail, time samples are dimensionless and consequently the reported amplitude is expressed in dB. The 0 dB reference corresponds, in the time domain, to the maximum coordinate (x and y, respectively) evaluated after the high-pass filter stage.

### 4.3. Stop Movement Test

As described in Section 2, to analyze the effect of stopping a movement, the robotic arm was programmed to take a different trajectory over time. In particular, the trajectory was divided into a locomotion part and a stop part. During the locomotion, up to about 3 s, a linear movement with an overlapping oscillation equal to 2 Hz was performed; subsequently, in the stop part (up to about 8.5 s) the arm stopped moving and an oscillation equal to 5 Hz was introduced. The choice of the two frequencies to be used complied with [38]. Figure 6 shows the acquired trajectories for all of the 10 repetitive stop movement tests.

Like the tremor test, the original trajectory was filtered with a high-pass IIR eighth order filter with a cut-off frequency of 1 Hz, and subsequently a DFT with a rectangular window was applied. Figure 7 shows the comparison between the spectrum of the original and the filtered trajectory.

All spectra of the filtered signals feature peaks at 2 Hz and 5 Hz, the nominal frequencies simulated with the servo motor. In both cases, these peaks represent the tones with the largest amplitude in the spectrum and can be easily identified. The 5 Hz tone is the largest for the x coordinate, while for the y coordinate the 2 Hz tone is the largest.

To observe the temporal and frequency evolution of the servo–motor oscillation intensity, Figure 8 shows the spectrograms of the filtered signals obtained by means of DFT on moving windows, being 30% wide (with respect to the total number of samples) and 50% overlapped. A frequency change occurring about time *t* = 3 s can be observed, both for the x and y signals; in particular, before *t* = 3 s, the 2 Hz component is present and, subsequently, until the end of the test, the 5 Hz one is the only component present.

### 4.4. Tapping Test

By mounting a TX on the tip of the index finger, we used the MPS to track the finger movement while performing finger tapping (FT) tests. The subjects involved in these FT tests were two of the authors. In the first test series, they moved their fingers at an almost constant rate, as any healthy control subject would in a real-life clinical test. In the second experiment, they performed the test by randomly slowing down the movement of the fingers, and making some short pauses, i.e., trying to replicate the movement features expected in a real parkinsonian patient. We refer to the second case as disease, even if it is just a simulation. Each FT test lasted for a few seconds, resulting in about 10–20 taps for each take. Note that both involved subjects are healthy individuals, and the presented activity was not a clinical study, but aims at proving that the MPS readings ability to correctly identify and classify the characteristic features of movements performed during an FT test. Consequently, these results provide foundations for future developments, where the identified features may be used to develop real clinical tests employing the MPS readings.

As an example, Figure 9 (top) shows the z coordinate of the index finger tip during a single FT take, for both the no-disease and disease cases. Since finger orientations were not significantly varied during the test, only one coordinate (z specifically) was relevant for the characterization of the movement. The raw coordinate values measured in the MPS reference frame have been rescaled as $z_{sc}=(z_{raw}-z_{min})/\left(z_{max}-z_{min}\right)$ so as to have a movement amplitude always normalized between 0 and 1, in order to extract features that are not dependent on the particular FT take, or on the subject being examined. Figure 9 (bottom) shows the velocities along the rescaled coordinate, calculated by finite differences.

Since, in the disease case, movements had some slower sections or even short pauses, one possible feature to investigate to discriminate between disease and no-disease cases is the fraction of sample points whose velocities are below a properly chosen threshold. After trying different threshold values on our dataset, a clear distinction was not obtained. This finding is consistent with the results presented in [29], hence, we decided not to investigate this feature further.

To identify other features, we looked at the frequency spectra of the *z* coordinate values for each FT take. Figure 10 shows a superposition of all the spectra of the dataset, together with their average values. There are 17 no-disease spectra and 20 disease spectra, including both test subjects. No low-frequency cut-off has been applied to these spectra.

The amplitudes are expressed in logarithmic units according to the following equation:(4)$$Amplitude\;\left[dB\right]=20\log_{10}z_{sc}$$

The 0 dB reference is obtained when $z_{sc}=1$. Even if the spectra are relative to two different persons, they overlap quite well, and a distinctive trend is observed. Disease and no-disease spectra are used as a heuristic observation, in order to identify distinctive features to be used in a classification scheme whose accuracy is studied in the following section. In order to immediately evaluate the statistical significance of the spectra, we applied a two-sample *t*-test. Let us consider the vectors of amplitude values falling in the i-th frequency bin, for both the disease and no-disease case; let us call them $A_{d}^{i}$ and $A_{nd}^{i}$, respectively. The *t*-test estimates the probability of the null hypothesis that both $A_{d}^{i}$ and $A_{nd}^{i}$ come from the same distribution. If the null hypothesis can be rejected for some of the frequency bins, with a significance level α=0.05, then the spectra can be considered statistically significantly able to discriminate between the disease and no-disease cases. It has been verified that the null hypothesis can indeed be rejected in the intervals 1.4–3.7 Hz, 4.6–5.5 Hz and 9.5–11.1 Hz.

Let us now identify some characteristic feature of the spectra. No-disease spectra have a distinctive peak around 5 Hz, with an observable first harmonic around 10 Hz, reflecting the good periodicity of movements in healthy subjects, as also verified in actual clinical tests [28]. Disease spectra are spread below 5 Hz, with no distinctive peaks. We fixed a 4 Hz threshold to discriminate between disease and no-disease cases. This threshold and the 5 Hz peak are compatible with data already shown in the literature, relative to real parkinsonian patients compared with healthy control subjects [31]. Hence, we defined two features to be associated with each FT take, *int1-4* and *int4-12*, i.e., the integral of the amplitude spectrum from 1 to 4 Hz and from 4 to 12 Hz, respectively, divided by the integral over the whole interval 0–31 Hz, and multiplied by 100, i.e., the weight in percentage of the 1–4 Hz and 4–12 Hz components over the whole spectrum. Such quantities are computed according to (5) and (6):(5)$$int1-4=100\int_{1Hz}^{4Hz}A\left(f\right)df/\int_{0Hz}^{31Hz}A\left(f\right)df$$
(6)$$int4-12=100\int_{4Hz}^{12Hz}A\left(f\right)df/\int_{0Hz}^{31Hz}A\left(f\right)df$$
where *A*(*f*) denotes the amplitude spectrum of the signal. It is expected for *int1-4* to be greater for disease cases, and, conversely, *int4-12* to be greater for no-disease cases. As shown in the scatter plot of the full dataset reported in Figure 11a, this expectation is quite well verified, with disease and no-disease cases forming two almost non-overlapping classes. Lastly, as a third feature, we used a parameter already defined in the literature, called the *finger tapping test score* (FTTS) [28]. The FTTS parameter is defined as:(7)FTTS=(PM−0.6)×amxfr,
where PM is a measure of the periodicity of the signal, while amxfr is a measure of the amplitude times the frequency of each tapping movement. To calculate PM, the signal is split into its *m* periods (*m* is different for each test). Local maxima $A_{i}=z_{sc}\left(k_{i}\right)$ are found, so as to mark the beginning of *i*-th period (*k* is the sample point index), then the following matrix is built, where each row is formed by the sample points of each period:$$Z=\left(\begin{array}{ccc} z_{sc}\left(k_{1}\right) & z_{sc}\left(k_{1}+1\right) & \begin{array}{cc} \cdots & z_{sc}\left(k_{2}-1\right) \end{array}\\ z_{sc}\left(k_{2}\right) & z_{sc}\left(k_{2}+1\right) & \begin{array}{cc} \cdots & z_{sc}\left(k_{3}-1\right) \end{array}\\ \begin{array}{c} \vdots \\ z_{sc}\left(k_{m}\right) \end{array} & \begin{array}{c} \vdots \\ z_{sc}\left(k_{m}+1\right) \end{array} & \begin{array}{c} \begin{array}{cc} \vdots & \;\;\;\;\;\;\;\;\;\vdots \;\;\;\;\;\;\;\; \end{array}\\ \begin{array}{cc} \cdots & z_{sc}\left(k_{m}+n\right) \end{array} \end{array} \end{array}\right)$$

Since the number of sample points *n* is different for each period, its median value is taken, and then each period is resampled by linear interpolation, to have the same number of elements in each row. The singular value decomposition (SVD) is then performed, as $Z=U\Sigma V^{†}$, where Σ is a diagonal matrix whose elements are the singular values $\sigma_{i}$, while U and V are semi-unitary matrices. For a perfectly periodic signal, only $\sigma_{1}$ would be a non-zero value [28]. Then, PM is defined as:(8)$$PM=\frac{\sigma_{1}^{2}}{\sum_{i}\sigma_{i}^{2}}.$$

For a perfectly periodic signal, PM=1 [28]. In general, PM is expected to be greater in no-disease than in disease tests.

As for amxfr, it is defined as:(9)$$amxfr=\frac{1}{n}\overset{n}{\underset{i=1}{\sum }}\frac{A_{i}}{T_{i}},$$
where $A_{i}$ and $T_{i}$ are the maximum amplitude and the time duration of each period. Greater movement amplitude and frequency, i.e., greater amxfr, are expected in no-disease cases.

Figure 11 shows the scatter plots for the complete dataset of all the features identified (*int1-4, int4-12,* FTTS). No-disease and disease cases are clearly split into almost non-overlapping classes. To estimate the accuracy of this classification, we applied the k-nearest neighbor (k-NN) algorithm, by using the classification learner app included in MATLAB. The number of neighbors was chosen to be 10. In order to prevent overfitting of the training data, MATLAB applies an *n*-fold cross-validation scheme, i.e., the dataset is randomly split into *n* subsets; *n-1* subsets are used to train the model, and one subset to test it; the procedure is repeated *n* times using different subsets to test the model; the final accuracy is an average over all the tests. We set n=5. Using just *int1-4* and *int4-12*, the test accuracy is 91.9 %, while using either *int1-4* and FTTS, or *int4-12* and FTTS, or all three features together, the final accuracy is always 94.6 %.

## 5. Conclusions

A tracking system based on magnetic localization was proposed and tested in this paper to track the movement of a robotic arm inside a limited cubic domain. Such activity is aimed at demonstrating the capability of the system to provide information about light symptoms associated with Parkinson’s disease. Unlike other available systems, which are based on inertial sensors or cameras, it tracks data with high spatial and temporal resolution, with respect to a fixed reference frame, without the drift caused by the integrative nature of inertial sensors, and it is operational even in the presence of obstructions that affect camera-based systems. Furthermore, our system is characterized by a much lower-cost, and it has an easier installation. Different types of tremors and trajectories were emulated, and the system correctly recognized them by outputting information about the position vs. time in a 3D domain and vibration frequencies associated with the movement itself. For a particular test, namely the finger tapping test, a k-NN classifier was employed and its capability to discriminate a healthy movement from a non-healthy one achieved up to 94.6% accuracy. The developed system could easily be used in medical offices due to its portability, low cost, low power and fast response in terms of output availability. It could therefore help doctors in fast diagnoses of early-stage Parkinson’s patients, thus enhancing the probabilities to help them to live better with more effective medical treatments. The current status of the system proves its capability to follow movements and tremors that are typically involved in Parkinson’s disease. In the near future, since the system’s performance with reference movements has been assessed, it can be subjected to an extensive experimental campaign where volunteers can be involved, and their movements evaluated and performance compared to the current controlled case.

## Figures and Tables

**Figure 1 sensors-21-04196-f001:**
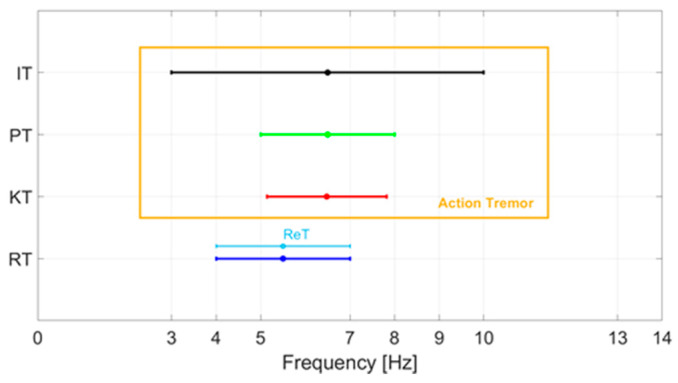
Characteristic frequencies of the various tremors of PD: rest tremor (RT), kinetic tremor (KT), postural tremor (PT), intentional tremor (IT), re-emerging tremor (ReT).

**Figure 2 sensors-21-04196-f002:**
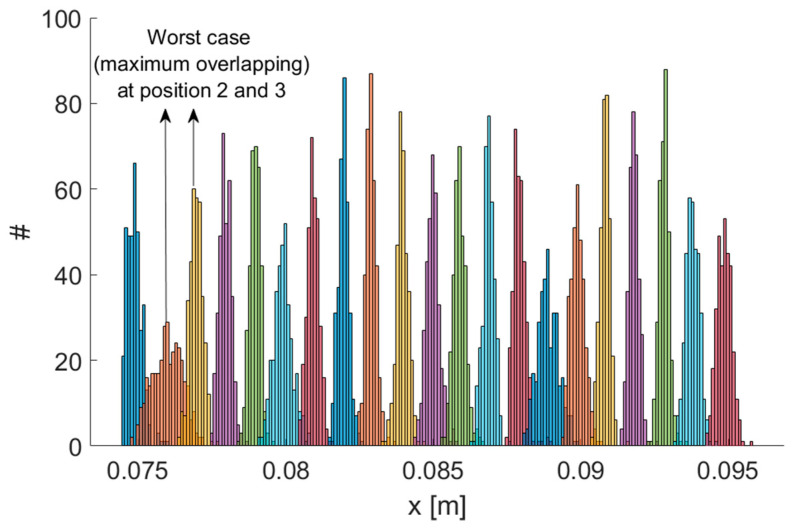
Testing the resolution of the MPS. Repeated measurement results on a set of positions spaced 1 mm. Maximum overlapping has been obtained for positions 2 and 3.

**Figure 3 sensors-21-04196-f003:**
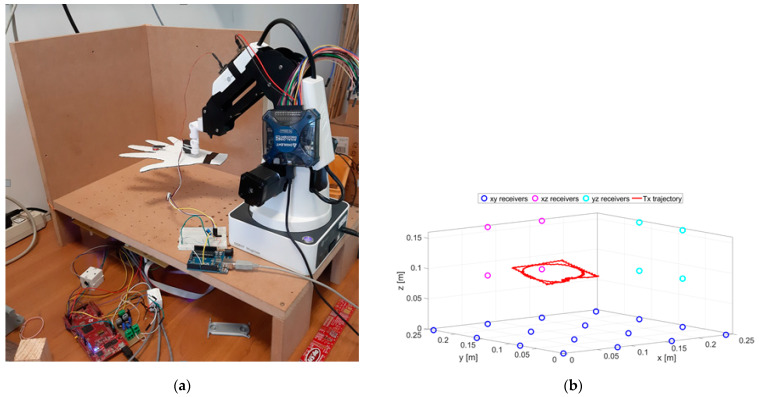
A picture of the experimental setup (**a**) and schematic representation of the RX positions and an example of the TX trajectory (**b**).

**Figure 4 sensors-21-04196-f004:**
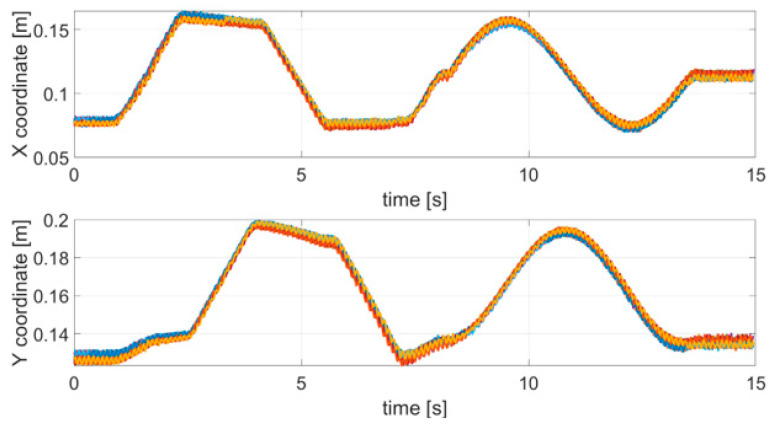
Acquired trajectories (x and y coordinates) for each tremor test carried out.

**Figure 5 sensors-21-04196-f005:**
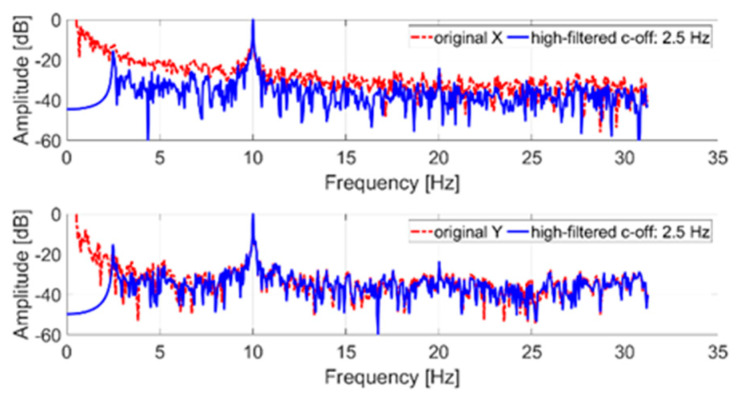
Frequency spectrum of an acquired tremor trajectory (x and y coordinates); in blue, the original signal; in red, the filtered signal.

**Figure 6 sensors-21-04196-f006:**
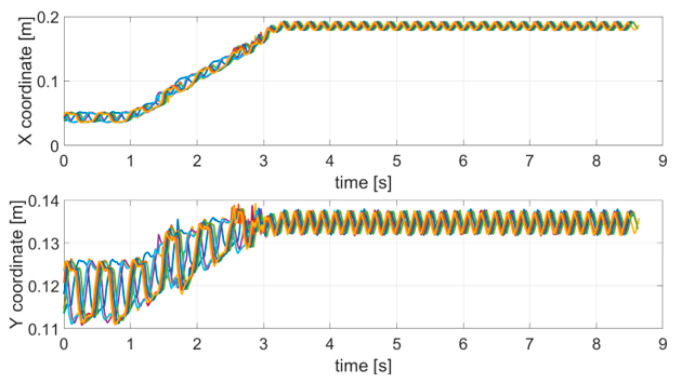
Acquired trajectories (x and y coordinates) for each SM test carried out.

**Figure 7 sensors-21-04196-f007:**
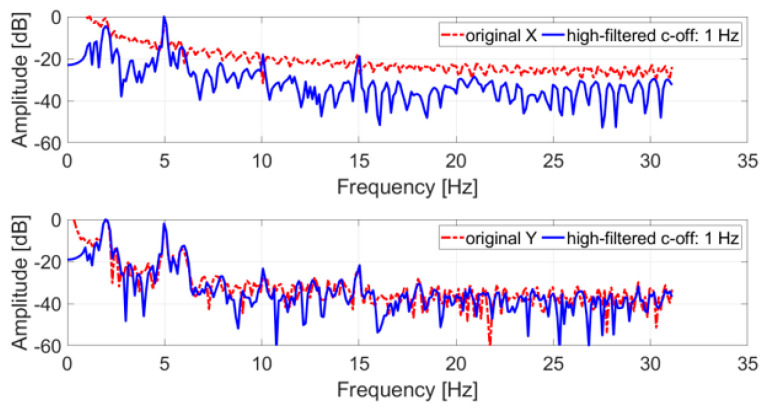
Frequency spectrum of an acquired SM test trajectory (x and y coordinates); in blue, the original signal; in red, the filtered signal.

**Figure 8 sensors-21-04196-f008:**
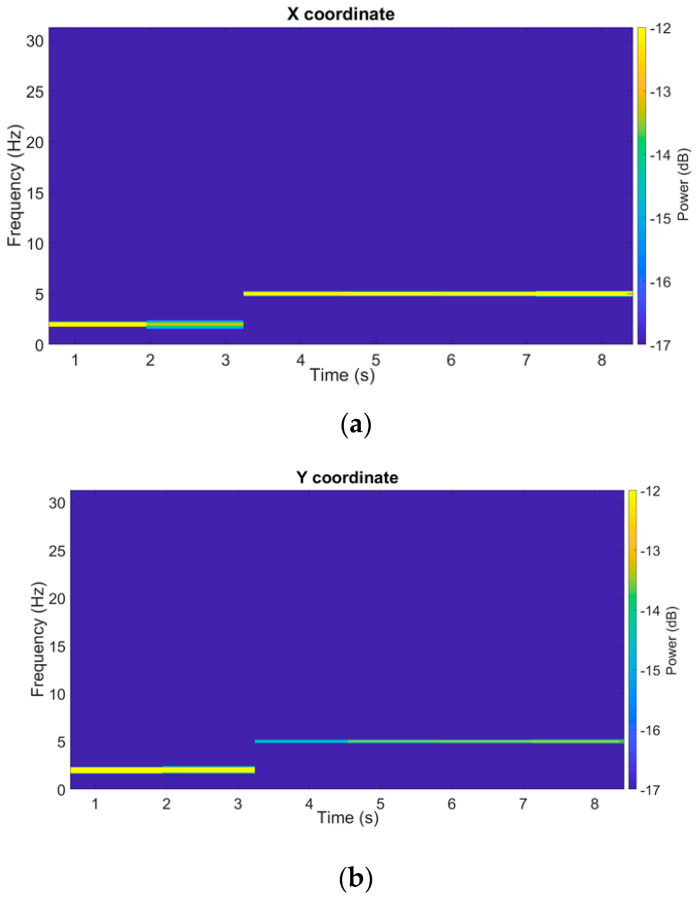
Spectrograms of a filtered SM test trajectory (x and y coordinates).

**Figure 9 sensors-21-04196-f009:**
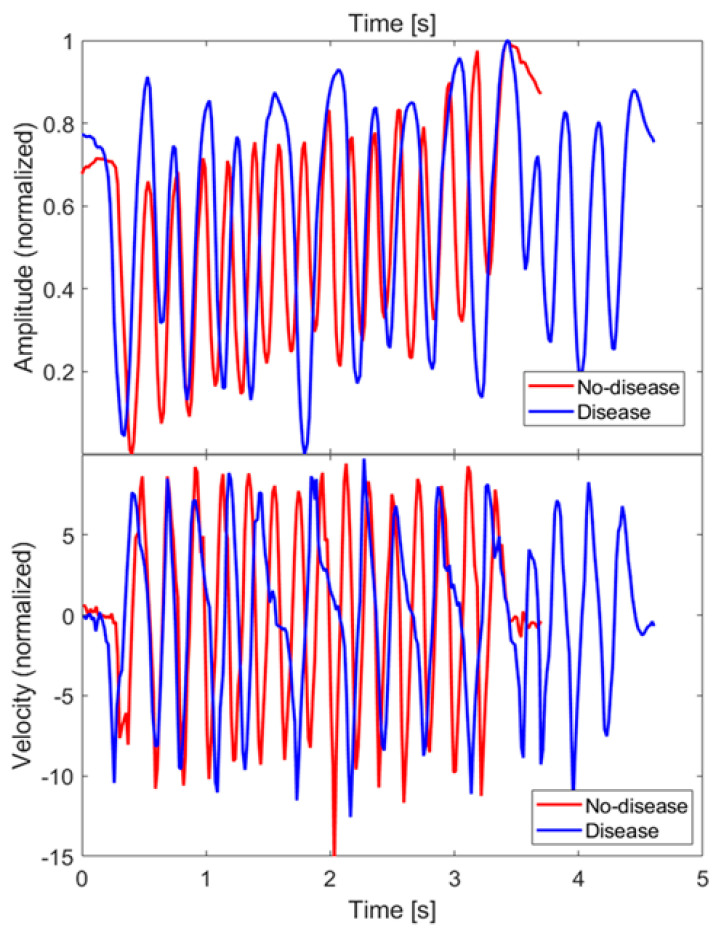
Amplitude ($z_{sc}$) and velocity of the FT of the index fingertip measured along z direction, for both the no-disease and disease cases. Velocity has been calculated by finite differences.

**Figure 10 sensors-21-04196-f010:**
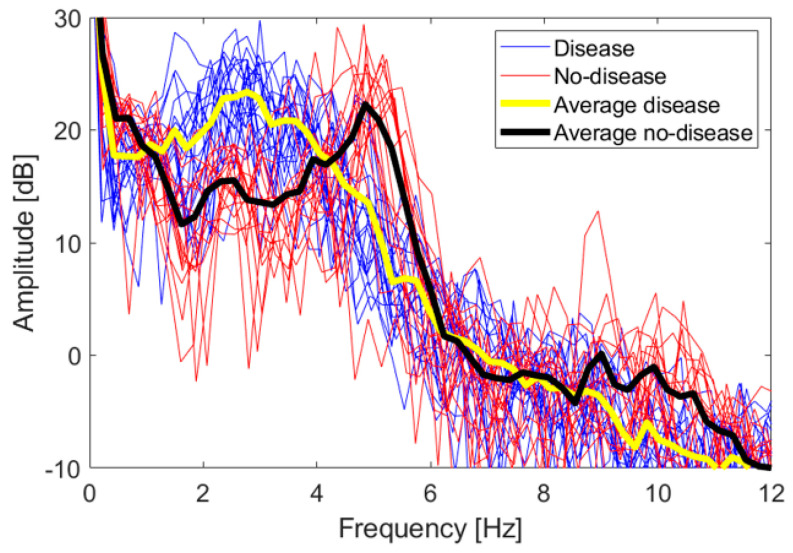
Frequency spectra of the z coordinate for both the disease and no-disease cases. The superposition of all spectra of each single FT take is shown, together with their average trend.

**Figure 11 sensors-21-04196-f011:**
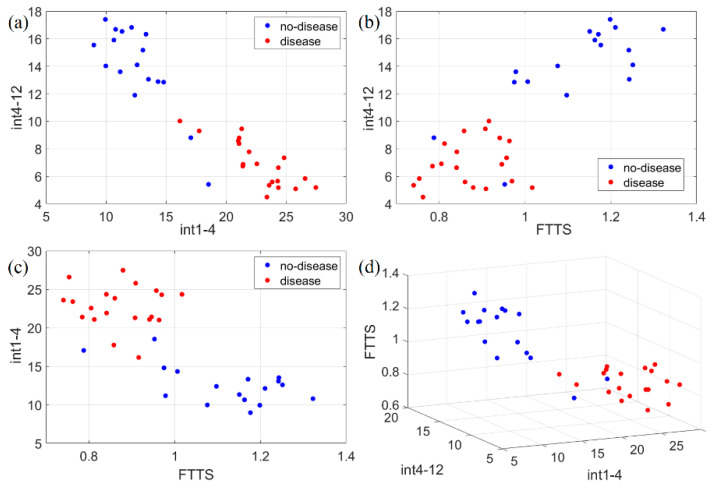
Scatter plots of the three identified movement features (*int1-4*, *int4-12*, FTTS). Such features have been plotted in 2D and 3D to check separability in the following ways: (**a**) 2D representation of *int1-4* vs. *int4-12*; (**b**) 2D representation of FTTS vs. *int4-12*; (**c**) 2D representation of FTTS vs. *int1-4*; (**d**) 3D representation of the three adopted features for classification. Disease and no-disease cases are well separated, forming almost non-overlapping classes with respect to each feature.

**Table 1 sensors-21-04196-t001:** Summary of the main features of the MPS.

${\bar{\sigma }}_{r}$ [mm]	3.5 ± 1.9
${\bar{\sigma }}_{\theta }$ [°]	2.6 ± 1.5
Meas. rate [Sa/s]	62
$\bar{SNR}$ [dB]	20 ± 13
TX freqs [kHz]	176–186

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
