# Peer review of "Parkinson’s Disease Patient Monitoring: A Real-Time Tracking and Tremor Detection System Based on Magnetic Measurements†"

_sensors, 2021, doi:10.3390/s21124196_

Round 1

Reviewer 1 Report

The paper proposes a magnetic tracking system able to collect information about translational and vibrational movements of the robotic hand that simulates the hand tremor of the people with Parkinson’s disease for patient monitoring.

Major comments:

  1. The introduction section discusses various methods for PD diagnostics but surprisingly misses one of the most prevalent method, which is based on the speech analysis, see for example, “Detecting parkinson's disease with sustained phonation and speech signals using machine learning techniques”, and “Parkinson’s Disease Diagnosis in Cepstral Domain Using MFCC and Dimensionality Reduction with SVM Classifier”.
  2. What is the spatial resolution of the proposed system?
  3. You used the k-nearest neighbors’ algorithm (k-NN) for classification. KNN is of course a machine learning algorithm. How would you motivate the use of a machine learning algorithm in this study considering the objections raised against the use of machine learning techniques in Lines 76-83? Isn’t your solution too specific and the outcomes can not be generalized?
  4. The proposed system is claimed to be “a scalable and transportable tracking and monitoring system” (Lines 84-85). What is the size and weight of the developed system? How does it compare to smartphone based systems for evaluating Parkinson’s and other central nervous system motor disorder symptoms? (see, for example, “A Smartphone Application for Automated Decision Support in Cognitive Task Based Evaluation of Central Nervous System Motor Disorders”).
  5. In overall, the evaluation of the system is provided in the Introduction section of the paper, which is bad from the structural point of presentation. Move the evaluation of the proposed system towards the end of the paper and place it in the evaluation of discussion section.
  6. The sampling rate of your system (10Hz) is not high as claimed, as other systems for tremor evaluation work at a much higher sample rate, for example, 100 Hz (see “Quantitative Assessment of Parkinsonian Tremor Based on an Inertial Measurement Unit”).
  7. Is there a statistically significant difference between no-disease spectra and disease spectra? The statistical analysis of the results must be performed.
  8. What is the correlation of the simulated signals with real Parkinson’s patients’ tremor signals? Did you perform the validation of your signal generation system?
  9. The study has no practical value as “No human patients are involved in the tests, and most cases are emulated by means of a robotic arm, suitably driven to perform required tasks”. Only the studies with human subjects can be used to support the diagnostics oriented claims and findings. I suggest to remove all claims regarding the diagnostical potential of the developed system.  

Minor comments:

  1. Avoid mass-citations such as “[2]–[7]”, but rather use each reference to support a separate claim.
  2. Line 66: “Reviews such as [12]” – plural noun suggests the existence of many studies supporting this claim. So more references should be provided to support it.
  3. Table 1: the caption is missing.
  4. Figure 2b: the simulation produces tremors in 2D plane only, which does not correspond to real tremor that is 3D.
  5. Lines 275-276: “The tremor component (10 Hz) introduced through the servomotor, is clearly identifiable on both axes.” – What do you do to remove the 10Hz component. Clearly, you should apply the notch filter.
  6. Figure 5: the figures do not provide much information. What should we see? I suggest to remove.
  7. Equations (5)-(6): the equations are wrong since you have removed all signals below 2.5Hz using a high-pass filter.
  8. Figure 11: explain the subplots in the caption. The caption claims three features, but I see four plots.

Reviewer 2 Report

The paper presents a novel method to assess patient with Parkinson's disease. The idea itself is impressive. However, there are several problems around the test protocol which needs to be addressed. 

1)  The paper used magnetic measurement for trajectory tracking as the test objective. However, it is vague why the trajectory tracking is important for assessment of Parkinson's disease?

2) Considering the system mentioned in the paper is fixed in the environment, Why not use camera or motion capture system? Even if the authors think the advantage of the proposed system is robust to obscure objects. However, why is it necessary to pay attention to this circumstance for assessment? 

3) In the test, why the authors not recruited real patients for the validation of the system? Is there any evidence to support that the robotic arm can simulate real Parkinson's disease patients?

Round 2

Reviewer 1 Report

The authors have addressed all my comments and improved the paper accordingly.

Author Response

Thank your positive evalution!

Reviewer 2 Report

The authors please mention how they modified the paper or edited the text in the response letter. 
